# Study on Deterioration Characteristics of Uniaxial Compression Performance and Microstructure Changes of Red-Bed Mudstone during Gaseous Water Sorption

**Hongbing Zhu [1,2], Zhenghao Fu [1,2], Fei Yu [3,\*] and Sai Li [4]**

[1]  School of Urban Construction, Wuhan University of Science and Technology, Wuhan 430065, China
[2]  Institute of High Performance Engineering Structure, Wuhan University of Science and Technology, Wuhan 430065, China
[3]  State Key Laboratory of Geomechanics and Geotechnical Engineering, Institute of Rock and Soil Mechanics, Chinese Academy of Sciences, Wuhan 430071, China
[4]  School of Urban Construction, Wuchang Institute of Technology, Wuhan 430065, China
\*  Correspondence: fyu@whrsm.ac.cn

**Abstract:** Previously conducted studies have established that gaseous water sorption of mudstone is widespread in nature. The deterioration of its uniaxial compression properties during gaseous water sorption can cause engineering problems. However, related studies were currently in the initial stage of this research direction. On the one hand, there were few studies on the deterioration characteristics of the uniaxial compression properties of mudstone in this process. The results might not be applicable to all projects. On the other hand, its microstructure changes in this process were unclear. Therefore, to obtain the deterioration characteristics of uniaxial compressive performance during gaseous water sorption for offering scientific reference to the geotechnical engineering of mudstone in the central Sichuan region of China, red-bed mudstone was used as a research material. A swelling test and uniaxial compression tests were carried out. To clarify microstructure changes for advancing the depth of research on the effects of gaseous water on mudstone, scanning electron microscopy (SEM) tests were performed. As a result of this study, formulas were first established that could correctly characterize the deterioration of uniaxial compressive strength (UCS) and elastic modulus when the moisture absorption rate increased. Secondly, the dependence was obtained, which was the relationship between both the UCS and elastic modulus and moisture absorption time. Finally, microstructure changes were revealed during gaseous water sorption.

**Keywords:** red-bed mudstone; gaseous water sorption; uniaxial compressive strength (UCS); elastic modulus; microstructure; deterioration characteristics; swelling test; uniaxial compression test; scanning electron microscopy (SEM); geotechnical engineering





## 1. Introduction

Mudstone contains hydrophilic clay minerals such as montmorillonite and illite. This means it is extremely sensitive to environmental factors such as temperature, humidity, stress, and groundwater [1,2]. Especially when mudstone is exposed to water or when the external humidity rises, its clay minerals will absorb moisture and swell, which leads to increased internal stresses and microstructural damage in the rock. The macroscopic manifestations are rock swelling and strength reduction [3–5]. Around the world, the weak water stability of mudstone had caused many engineering problems, such as arching on high-speed railway foundations [6], tunnel collapse [7], and landslides [8]. Therefore, it is an attractive topic for engineering construction to study the water effect on mudstone.

In recent years, there have been numerous research studies on the effect of water on mudstone. Through indoor swelling tests, Zhong [9] qualitatively pointed out that the swelling of mudstone can be divided into three stages, such as the rapid swelling

stage, the decelerated swelling stage, and the swelling stabilization stage. the Initial water content [10], the type of water solution [11], and water pressure [12] have been shown to be water-related influencing factors on mudstone swelling. With the deepening of research, Al-Rawas [13] and Chen [14] concluded that clay mineral content is the fundamental factor affecting water absorption and the swelling of mudstone. Besides, Chen pointed out that the uneven distribution of clay minerals in space would lead to the uneven expansion of the mudstone and argued that this uneven expansion would cause new fractures in the rock. The swelling of mudstone is a combination of fracture expansion and matrix swelling [5,15]. The former destroys the structural integrity, and the latter changes the properties of the rock material, which may influence the mechanical properties of the mudstone. As one of the key geomechanical properties of rocks, the uniaxial compressive strength (UCS) is closely related to practical engineering and numerical simulation. For instance, to reduce foundation settlement, UCS is raised by compacting the soil [16]. Rock with different UCS can affect the piling process [17]. The behavior of rock mass is predicted during salt mining [18]. In the numerical simulation, the calibration of the micro parameters of the established model was achieved by the measured uniaxial compressive property parameters (UCS and elastic modulus) [19], etc. It is of importance in civil engineering, mining, geotechnical, and infrastructure projects [20,21]. Therefore, it is valuable to clarify the changes in the UCS of mudstone during the moisture absorption process. Numerous studies have demonstrated that water can reduce the energy required to produce cracks and increase the length of internal cracks [22,23]. The UCS and stiffness of the mudstone decreased with increasing moisture content [24–27]. In addition, Liu [3] established the relationship between moisture content and different mechanical parameters (elastic modulus and UCS) by numerous uniaxial compression tests. Through the scanning electron microscope (SEM) test, he pointed out that the pores of the microstructure increased significantly after wetting and the structural integrity was broken. This is the reason for the decrease in the UCS of the mudstone. As a result, the impact of water absorption on mudstone was systematically revealed. However, all the above studies used liquid water adsorption to investigate the effect of moisture absorption on mudstone. The findings may not be applicable to all engineering.

In nature, the occurrence environment of mudstone is very complex. It has a variety of contact modes with water, and not only liquid water absorption occurs, but even gaseous water absorption is carried out in high humidity environments [28–32]. The crack growth velocity of rock in water or an aqueous solution is much higher than that in high-humidity environments [28]. Moreover, the swelling law of mudstone during gaseous water sorption is completely different from that of liquid water sorption [29]. These indicate that the effect of gaseous water sorption on the mechanical parameters of mudstone may be different from that of liquid water sorption. Zhang [31] and Fu [32] considered that studying the deterioration characteristics of rocks in high-humidity environments is important for the design and maintenance of unconventional engineering and has the value of in-depth research. In recent years, there have been few research studies reflecting the deterioration of mechanical properties of mudstone during gaseous water sorption. Vales [33] and Cherblanc [34] pointed out that the softening characteristics of clay rocks are obvious, and their UCS and elastic modulus are significantly reduced in this process. However, the findings did not involve the effect of gaseous water on the microstructure of rock samples. Although Houben [35] observed by environmental scanning electron microscope (ESEM) tests that cracks of mudstone developed and closed in the process of gaseous water sorption due to the inhomogeneous expansion of clay minerals, the primary and secondary relationships between the expansion and closure of cracks were not specified. In brief, the microstructure changes in mudstone during gaseous water sorption are not clearly defined.

Red layer is a kind of semi-consolidated rock; its strength is between the loose layer and hard rock, controlled by the environment humidity of the fault zone in the distribution area. In particular, the red-bed mudstone distributed in the Sichuan Basin was exposed to an environment with high humidity for a long time due to the influence of the Sichuan

Basin climate. It had caused a variety of engineering diseases in the central Sichuan region of China [6,9,14,29]. Since the scientific understanding of the degradation of its mechanical properties during gaseous water sorption was lacking, this has caused great obstacles to the design and construction of geotechnical engineering in the high humidity environment of the area. In addition, the relationship between the water content and UCS is strongly influenced by the kind of rock. Negative exponential relationships [3,36] and negative linear relationships [37] have appeared in existing studies. Hence, it is also undesirable to directly refer to the relevant research results of other types of rocks to guide the actual project.

Analyzing the above, there are two issues that need to be studied urgently. On the one hand, the deterioration effect of mechanical properties brought by the moisture absorption of mudstone has caused a large number of engineering problems worldwide. It has high research value. Not only can the liquid water sorption deteriorate the mechanical properties of mudstone but the gaseous water sorption also can deteriorate its mechanical properties. However, the research on the deterioration of mechanical properties of mudstone during the gaseous hygroscopic process is limited in the existing studies. Especially, the changes in the microstructure during this process are not clear, which is not conducive to revealing the reasons for the deterioration of the mechanical properties of the mudstone. It needs to be investigated. On the other hand, the deterioration characteristics of the UCS of red-bed mudstone during gaseous water sorption are not known. Moreover, because the relationship between water content and UCS is strongly influenced by the kind of rock, previous relevant findings may not be referenced by actual engineering. This seriously hampers the design, construction, and maintenance of geotechnical engineering in the high-humidity environment in central Sichuan, China. Therefore, the purpose of this study is to clarify the deterioration characteristics of uniaxial compression performance and microstructure changes of red-bed mudstone during gaseous water sorption. The results of the study are hoped to complement studies on the microstructural changes of mudstone in this process and advance the depth of research on the effects of gaseous water on mudstone while providing a scientific reference for practical engineering. To achieve this, the following works were carried out. (i) Swelling tests were conducted to verify that a high humidity environment could affect the red-bed mudstone. (ii) Uniaxial compression tests were carried out to obtain the deterioration characteristics of uniaxial compressive performance during gaseous water sorption. (iii) SEM tests were performed to investigate microstructure changes in this process.

## 2. Basic Physical Parameters

### 2.1. The Stratigraphic Lithology and Climatic Characteristics of Sampling Points

The red-bed mudstone in central Sichuan, China was taken as the research object. The fresh mudstone was obtained from a tunnel in Neijiang City, Sichuan Province, located in the gentle low fold zone in central Sichuan, which was a monoclinic structure and nearly horizontal rock formation. The overlying slope residual (Q4dl + el) swelling soil was 0–2 m thick and the underlying bedrock was Jurassic Mesoproterozoic Upper Shaximiao Formation (J2s) mudstone interbedded with sandstone. The fresh mudstone excavated on site was complete in appearance, and its joints and cracks were not developed. It was brownish-red in color with a few parts of greenish-gray, and the surface had an oil-like substance distribution (see Figure 1). In addition, it had a medium–thick lamination with a certain swelling capacity. In the natural state, it was easy to weather and flake. It also had the behavior of softening and disintegration in water. For the fresh mudstone excavated, they were immediately sealed on site with cling film to prevent them from exchanging moisture with the air. The tunnel was in the Sichuan basin. It belonged to the subtropical monsoon climate with an annual rainfall of about 1000 mm. Although the liquid water in the tunnel was not obvious and no aggregated water was seen, the humidity in the middle of the tunnel was high. The relative humidity was at 80% for a long time and could reach about 95% in the evening. By measurements, the natural water content of the mudstone was 2.64~3.42% and its natural density was 2.56~2.60 g/cm$^3$.

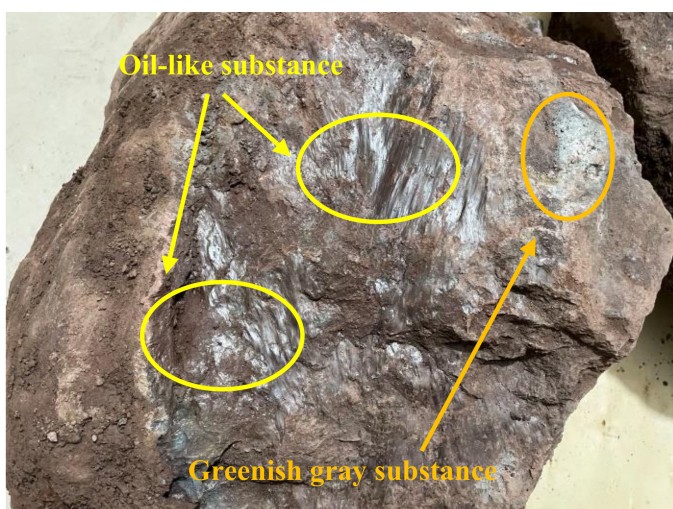

**Figure 1.** Excavated natural mudstone.

*2.2. Mineral Composition*

As a typical soft rock containing clay, the moisture absorption capacity of mudstone depends on its clay mineral content. Moreover, the relationship between them is positively correlated. To obtain the mineral composition of rocks, especially the content of hydrophilic clay minerals such as montmorillonite, illite, and kaolinite, 10–20 g of samples was selected, dried, ground, and sieved at 0.05 mm. Then, X-ray diffraction (XRD) tests were performed. Table 1 suggests the results of the XRD mineral analysis. It indicates that the content of clay minerals was high and dominated by montmorillonite and illite. The content of the former ranged from 19.67% to 20.14% and that of the latter ranged from 14.54% to 15.36%. Thus, the red-bed mudstone used in this test had the material basis for water absorption and softening.

**Table 1.** The XRD results of mudstone.

| Specimen Number | Mineral Content (%) | | | | | | |
|---|---|---|---|---|---|---|---|
| | **Montmorillonite** | **Elysium** | **Quartz** | **Albite** | **Ridge Stone** | **Calcite** | **Hematite** |
| Specimen 1 | 19.67 | 15.36 | 35.32 | 18.11 | 5.12 | 4.75 | 1.66 |
| Specimen 2 | 20.14 | 14.54 | 38.63 | 20.54 | 4.88 | - | 1.27 |

**3. Design of the Test**

The red-bed mudstone has the characteristics of softening and disintegration in water and the initially hidden microcracks are developing within the rocks. The specimens were processed by using wire cutting machines because of the large disturbance caused by the dry drilling and water drilling methods. Referring to the previous research [3,29], the rocks were processed into two kinds of standard cylindrical samples (diameter was 50 mm and height was 100 mm. Diameter and height all were 50 mm). It needs to be mentioned that the non-parallelism between the top and bottom of the specimens did not exceed 0.02 mm. In addition, the end face was perpendicular to the axis and the maximum deviation was less than 0.2°. Eleven samples without oil-like substance and greenish gray substance on the surface were used for the test, including ten specimens for uniaxial compression tests and one specimen for the swelling test.

Since the purpose of the test was to reveal the variation pattern of uniaxial compressive properties of mudstone during gaseous water sorption, eleven specimens were dried in a temperature-controlled oven before the test. During the drying process, the temperature in the drying oven was set to 60 °C and the mass of the samples was weighed using a precision balance. When the change of their mass within 2 h was less than 0.01 g, it was

considered that the specimens did not contain water. Statistically, the average drying time of these samples was about 36 h. This can exclude the interference of initial moisture content. After the drying process, it was observed that there were no obvious cracks on the surface of the samples. Based on the objectives of this study, the experimental groups are shown in Table 2. Besides, it should be particularly clarified that five moisture absorption rates (it represented the degree of moisture absorption and was equal to the mass of water absorbed divided by the mass of the sample before moisture absorption) were only set, including 0% (drying sample), 1%, 2%, 3%, and 4%. There are two reasons. On the one hand, the pre-moisture absorption experiments displayed that the mudstone could not reach saturation condition by gaseous water sorption even at a 99% RH environment. Its maximum moisture absorption rate was 4.10%. On the other hand, published research has shown [3,4] that the early stage of moisture absorption is the most obvious stage of its mechanical properties decrease. Moreover, the effect of water has the greatest influence and deserves to be focused on.

**Table 2.** Test groups.

| Test | Specimen Number | Number of Specimens | Specimen Size | Purpose |
|---|---|---|---|---|
| Gaseous water sorption uniaxial compression test | KY0-1, KY0-2 | 10 | Height × diameter: 100 mm × 50 mm | 0% moisture absorption rate uniaxial compression test |
| | KY1-1, KY1-2 | | | 1% moisture absorption rate uniaxial compression test |
| | KY2-1, KY2-2 | | | 2% moisture absorption rate uniaxial compression test |
| | KY3-1, KY3-2 | | | 3% moisture absorption rate uniaxial compression test |
| | KY4-1, KY4-2 | | | 4% moisture absorption rate uniaxial compression test |
| Gaseous water sorption swelling test | PZ1 | 1 | Height × diameter: 50 mm × 50 mm | Measurement of swelling rate |

The processing procedure of the samples is shown in Figure 2. First, the swelling test was performed on one drying rock to know its swelling characteristics. On this basis, the impact of gaseous water sorption on the red-bed mudstone was initially determined. Then, the other eight specimens were put into closed containers for gaseous water sorption and the weight was weighed every 6 h to calculate the moisture absorption rate. When it reached the planned moisture absorption rate (1%, 2%, 3%, 4%), the samples were removed, and the duration of the moisture absorption was recorded. Finally, they were placed in a pressure tester for uniaxial compression testing (see Figure 3b).

In the swelling test, the specimen was placed in a customized swelling test box of gaseous water sorption (see Figure 3a), whose dimensions were 52 cm × 40 cm × 30 cm (length × width × height). With the help of a micrometer, the expansion of the sample was measured. In addition, the experiments were all conducted by saturated $K_2SO_4$ solution to create a stable high humidity environment [29] (RH99%), in which the mudstone was allowed to absorb gaseous water. As displayed in Figure 3b, a digitally controlled electro-hydraulic servo testing machine (its loading rate could be controlled from 0.0001 mm/s to 1.0 mm/s) was used for uniaxial compression tests. The rock sample was placed on the cylindrical bottom plate of the machine and the upper top surface of the specimen was in contact with the counterforce equipment of the machine. During the test, the bottom plate was kept stable, and loading was achieved by applying vertical downward displacement to the counterforce equipment. Moreover, the loading rate was set at 0.002 mm/s.

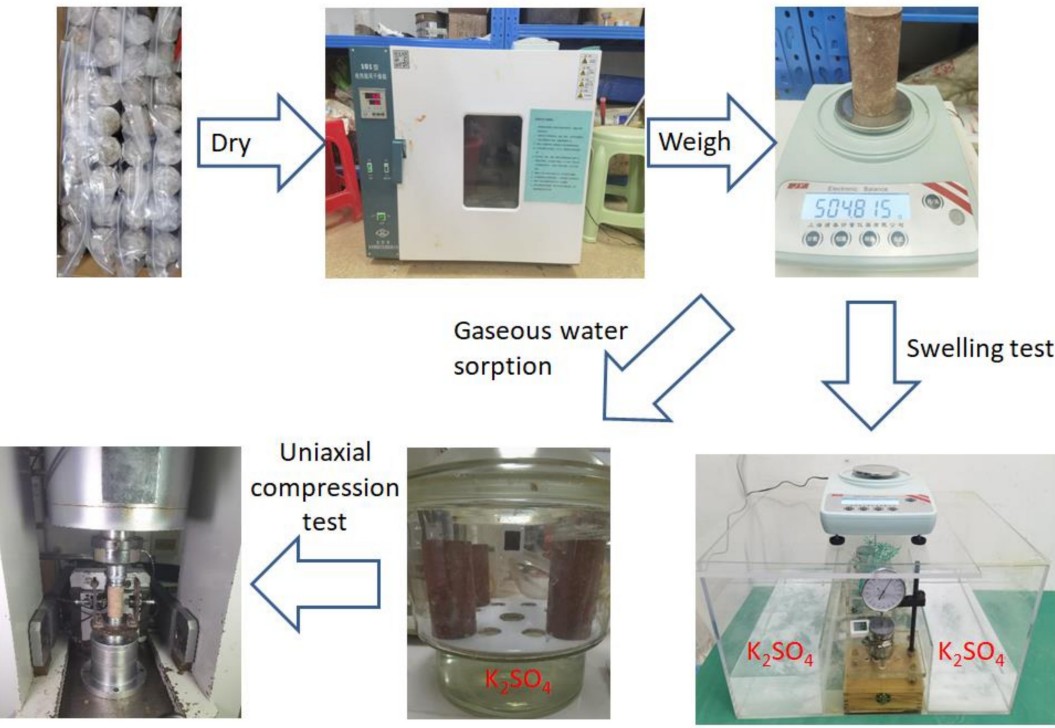

**Figure 2.** The processing of tests.

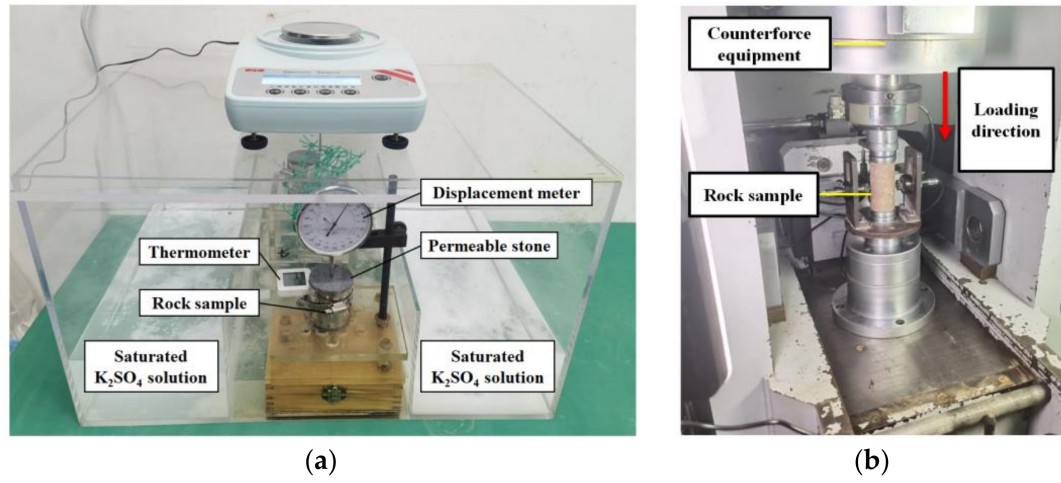

**Figure 3.** Test device: (**a**) The swelling test; (**b**) The uniaxial compression test.

## 4. The Results and Discussion

### 4.1. The Swelling Test

Because the moisture-absorbing swelling of mudstone is a combination of matrix swelling and crack expansion, it can reflect the structural deterioration to some extent. The swelling test was performed on the mudstone sample and Figure 4 showed the variation pattern of the swelling rate with time. As can be seen in Figure 4, the swelling process could be divided into four stages. The first stage was the rapid swelling stage, in which the specimen swelled the fastest. The second stage was the intermittent swelling stage. In this process, swelling appeared with several interruptions (it showed a swelling stop for more than 2 h, which has been marked with a red circle in Figure 3). Moreover, the expansion speed rate increased and decreased without any obvious rules. The third stage was the speed reduction expansion stage. The swelling speed rate declined significantly, but the specimen continued to swell. The fourth stage was the stable stage, in which the swelling

speed rate of the sample was almost zero and its deformation was relatively stable. Overall, the swelling test was conducted for about 1800 h. The sample reached swelling stability at 1546 h after the start of the test, and it was with a significant time effect. Moreover, the final swelling rate was 0.406%, and the final moisture absorption rate was 4.05% (the moisture absorption rate was calculated by the following equation). This indicates that the swelling of mudstone in the gaseous water sorption mode was of long-term character.

$$w = \frac{m_2 - m_1}{m1} \times 100\% \tag{1}$$

where w indicates the moisture absorption rate (%), $m_2$ is the weight of the specimen after moisture absorption (g), and $m_1$ is the weight of the specimen before the test (g).

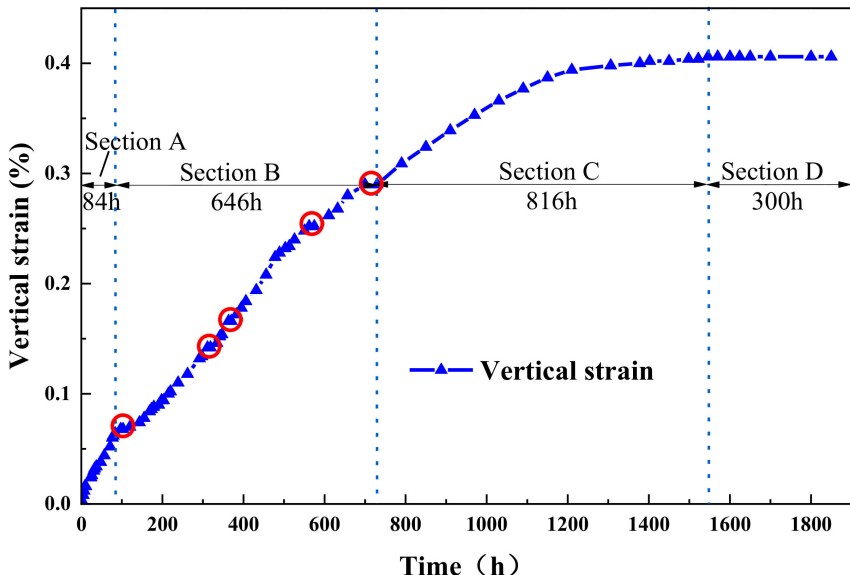

**Figure 4.** Swelling rate curve with time.

Combined with the results of the XRD test in Section 2, the clay minerals such as montmorillonite and illite of the specimen had a high content of about 35%. Further, according to existing studies [13,14], mudstone with high clay content is strongly influenced by water. Large moisture absorption and swelling should be produced in the moisture absorption process. However, the results of the swelling test differed significantly from those expected. The measured swelling rate and moisture absorption rate were small. The reason for this might be related to the pore structure of the rock. Since the permeability of mudstone is very poor, its pore structure acts as the main channel for water absorption and determines the water movement [38]. When it had low porosity, the number of clay minerals that could contact gaseous water was also low. This could cause the mudstone to have a small moisture absorption and swelling rate. In brief, when the specimen had fewer original pores, its swelling rate and moisture absorption rate were small during gaseous water sorption.

In fact, the swelling process of mudstone is a long-term complex physical-chemical reaction process under the combined effect of water and stress release [39]. Gaseous water sorption could lead to long-term swelling of red-bed mudstone. This suggested that certain physical-chemical reactions occurred within the specimen in the process of gaseous water sorption. Furthermore, this would negatively affect the uniaxial compressive properties of the mudstone and had a time-dependent behavior.

### 4.2. The Uniaxial Compression Test

(1)　Stress-strain curve

Figure 5 showed the stress-strain relationship for the KY0-2 specimen under uniaxial compression. This figure illustrated that the curve could be divided into four stages such as OA, AB, BC, and after C. The OA stage was the crack closure stage, in which the curve was up-concave, showing a typical nonlinear deformation. Moreover, the strain of this stage could indirectly reflect the information about the pores of the specimen [3,40]. It was small, about 0.0047. This indicated that the samples in the dried state had few microcracks and good structural integrity. It could also explain the small moisture absorption rate and swelling rate of the specimen during gaseous water sorption to a certain extent. The AB stage was the elastic deformation stage. Stress and strain exhibited a clear linear relationship with elastic deformation playing a dominant role. The existing study [40] showed that the deformation can be restored to near point A by unloading at this stage. During the first two stages of loading, no significant morphological changes were observed in the stressed specimens. The BC stage was the micro-crack propagation stage. First, the slope of the curve decreased significantly near point B, and an inflection point appeared with the increasing load. This indicated that a micro-crack in the specimen began to propagate. Then, the curve rose slowly until point C, where the specimen developed large cracks and was destroyed. In this stage, the test phenomenon was obvious. The sample developed cracks and they expanded rapidly. In addition, small rock fragments continuously fell off during this process (see Figure 5). After point C, the curve entered the post-peak failure stage. The sample had been destroyed. The stress fell sharply, and the device started to unload.

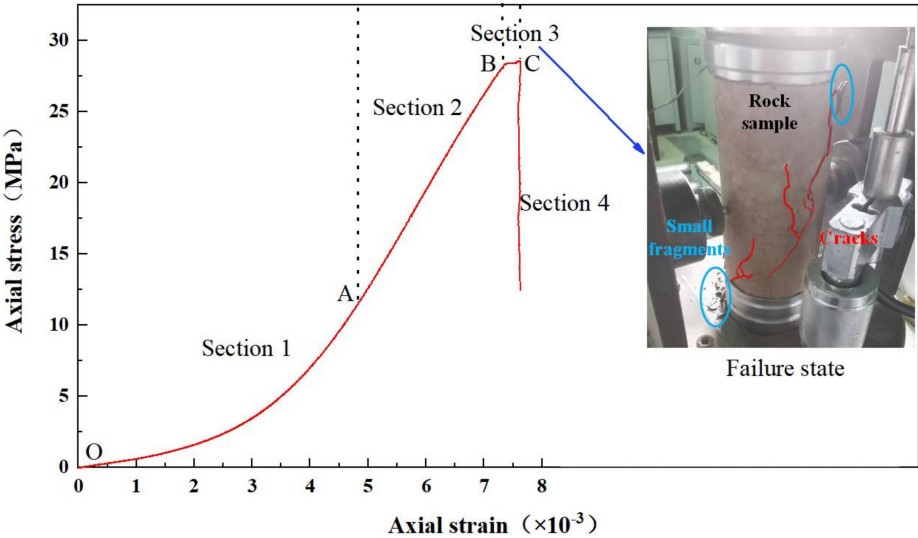

**Figure 5.** Stress-strain curve of the KY0-2 specimen.

(2)　Stress-strain curve characteristics of samples at different moisture absorption rates

Table 3 showed the compaction strain, elastic modulus, UCS, and failure strain of mudstone at different moisture absorption rates. Figure 6 illustrated the stress-strain curves (rising section) of the specimens at 0%, 1%, 2%, 3%, and 4% moisture absorption rates. Overall, the UCS of specimens decreased from 30.073 MPa to 5.870 MPa with the increase in the moisture absorption rate. Moreover, the failure strain rose from 0.007 to 0.009 in this process. This indicated that the compressive strength of the specimen declined, and the deformation capacity raised during gaseous water sorption. Significant softening of the mudstone occurred.

**Table 3.** Results of uniaxial compression test for specimens with different moisture absorption rates.

| Specimen Number | Water Absorption Rate (%) | | Compaction Strain ($10^{-3}$) | | Elastic Modulus (GPa) | | UCS (MPa) | | Failure Strain ($10^{-3}$) | |
|---|---|---|---|---|---|---|---|---|---|---|
| | Tested | Ave | Tested | Ave | Tested | Ave | Tested | Ave | Tested | Ave |
| KY0-1 | 0 | 0 | 3.54 | 4.15 | 6.887 | 6.927 | 30.073 | 29.333 | 7.32 | 7.47 |
| KY0-2 | 0 | | 4.76 | | 6.967 | | 28.593 | | 7.61 | |
| KY1-1 | 1 | 1 | 5.15 | 5.36 | 5.874 | 5.544 | 23.742 | 23.558 | 8.37 | 8.51 |
| KY1-2 | 1 | | 5.52 | | 5.213 | | 23.374 | | 8.64 | |
| KY2-1 | 2 | 2 | 5.41 | 5.53 | 3.486 | 3.757 | 16.745 | 17.686 | 9.18 | 8.92 |
| KY2-2 | 2 | | 5.64 | | 4.028 | | 18.626 | | 8.65 | |
| KY3-1 | 3 | 3 | 5.61 | 5.74 | 2.49 | 2.649 | 9.714 | 10.128 | 8.39 | 8.27 |
| KY3-2 | 3 | | 5.86 | | 2.807 | | 10.541 | | 8.15 | |
| KY4-1 | 4 | 4 | 6.1 | 5.71 | 1.218 | 1.123 | 5.870 | 6.487 | 9.51 | 9.37 |
| KY4-2 | 4 | | 5.51 | | 1.029 | | 7.103 | | 9.22 | |

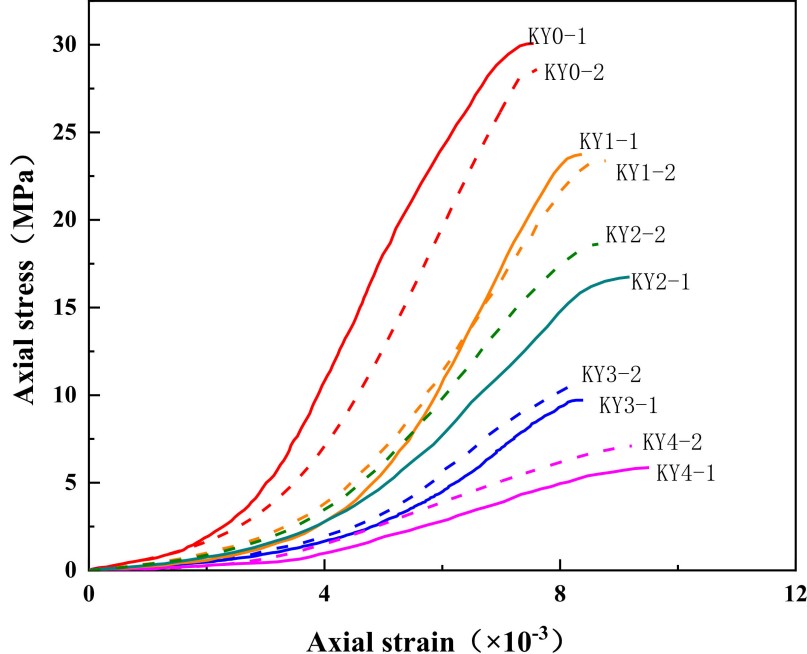

**Figure 6.** Stress-strain curves for specimens with different moisture absorption rates. note: The figure showed only the rising section of the stress-strain curve (the stage before the peak stress). The reasons for this were twofold. On the one hand, the decreasing section of the curve (the post-peak failure stage) had little influence on the purpose of the test. On the other hand, there were many curves in the graph. If all the descending sections of the curves were added, it would look like the information on the graph was cluttered and inconvenient to mark.

For the crack closure stage, the strain rose from 0.004 to 0.006 with the moisture absorption rate increasing, and their relationship was positively correlated. Liu [3] and Guo [40] believed that this is due to the expansion of the mudstone's own cracks caused by the process of moisture absorption. Therefore, it could be concluded that although the internal pores of the specimens expanded and closed during gaseous water sorption [35], the porosity of the mudstone was increasing in overall terms. Besides, the change of pores was dominated by expansion. In the elastic deformation stage, the slope of the curves declined when the moisture absorption rate rose. This suggested that gaseous water sorption could cause a decrease in the elastic modulus of the specimen. About the micro-crack propagation stage, the strain was the longest when the specimen reached a 4% moisture absorption rate. The specimens at 0% moisture absorption had the shortest strain.

Moreover, just after the inflection point, the specimens were destroyed. This indicated that gaseous water sorption led to an increase in the plasticity of the mudstone.

Therefore, in the process of gaseous water sorption, the elastic modulus and UCS of specimens declined, and their deformation capacity rose. Further, the softening characteristics of mudstone became more and more obvious.

(3)    Effect of moisture absorption rate on UCS and elastic modulus

It was observed that the specimens did not crack and disintegrate in the process of gaseous water sorption. Only a deepening of their color occurred. This might lead to an underestimation of the softening effect caused by gaseous water sorption. In contrast, from the above analysis, it was clear that it led to a significant drop in the UCS of the mudstone. Furthermore, compared to the drying sample, the maximum decrease in UCS was up to 80.5% when the moisture absorption rate of specimens was 4%. These were also exhibited in Figure 7. This could have a huge impact on the design, construction, and maintenance of the project. Consequently, it was meaningful to acquire the relationship between UCS and the moisture absorption rate.

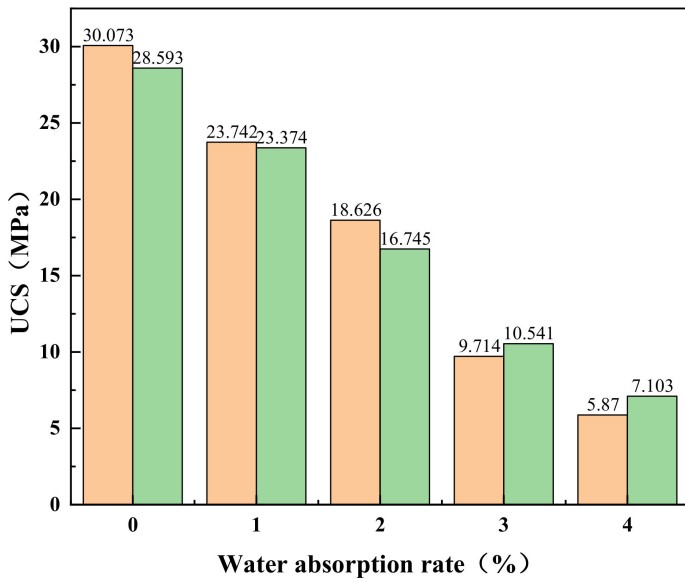

**Figure 7.** UCS with different moisture absorption rates.

In available studies, Liu, Yilmaz, and Yao used negative exponential relationship [3,36] and negative linear relationship [37] models to describe the relationship between UCS and moisture absorption rate, respectively. Moreover, the correlation was good. The details of the formula were displayed in Table 4. The differences in the relational models were not only caused by the different types of rocks, but also by the different range of independent variables w (moisture absorption rate). Observing the data curves of Liu and Yilmaz, it could be found that the UCS changed very significantly when the w varied in a lower range. However, while the w was at a high level, its effect on the UCS was not significant. Thus, if the set range of moisture absorption rate was low, it was also feasible to use a negative linear relationship model to describe the relationship between UCS and the moisture absorption rate in this range. Combining the existing outcomes and experimental data, a negative linear relationship was used to describe the relationship of the two. In addition, after fitting the data, it was found that the linear relationship was excellent (see Figure 8) with $R^2 = 0.98$ and the equation was given in Equation (2).

$$\sigma = 29.43 - 5.95w, \quad 0\% \leq w \leq 4\% \tag{2}$$

where $\sigma$ was UCS (MPa) and w was moisture absorption rate (%).

**Table 4.** Model of the relationship between UCS and moisture absorption in existing studies.

| Model | Relational Expression | Range of Moisture Absorption Rate ($w$) | a | b | c | $R^2$ | Existing Studies |
|---|---|---|---|---|---|---|---|
| Exponential | $\sigma = ae^{-bw} + c$ | $0\% \leq w \leq 6.2\%$ | 19.32 | 0.52 | 4.06 | 0.97 | Liu |
|  |  | $0\% \leq w \leq 8\%$ | 16.68 | 0.82 | 24 | 0.93 | Yilmaz |
| Linear | $\sigma = a - bw$ | $0\% \leq w \leq 5.5\%$ | 23.25 | 2.56 |  | 0.95 | Yao |

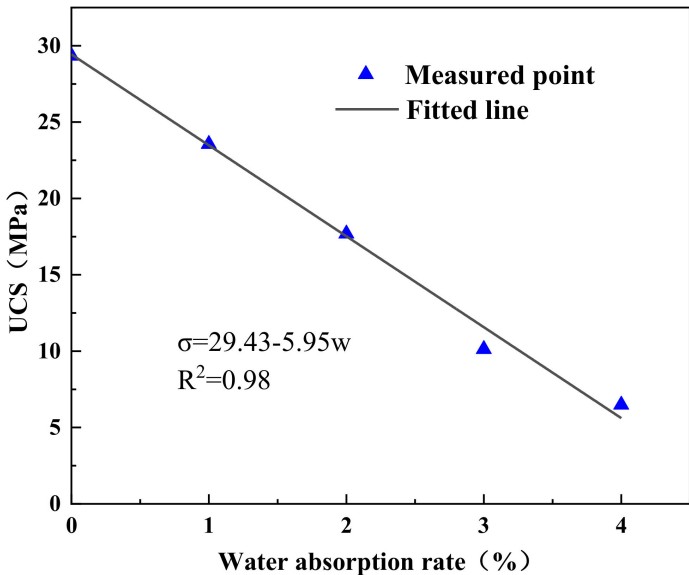

**Figure 8.** Relationship between UCS and moisture absorption rate.

The elastic modulus in this paper referred to the slope of the elastic deformation stage of the stress-strain curve. The relationship between the elastic modulus and moisture absorption rate was shown in Figure 9. With the increase in the latter, the former gradually decreased, up to 83.8%. Moreover, the linear relationship between the two was excellent and could be expressed by the following formula.

$$E = 6.90 - 1.45w, \ 0\% \leq w \leq 4\% \tag{3}$$

where E was the elastic modulus (GPa), and w was the moisture absorption rate (%).

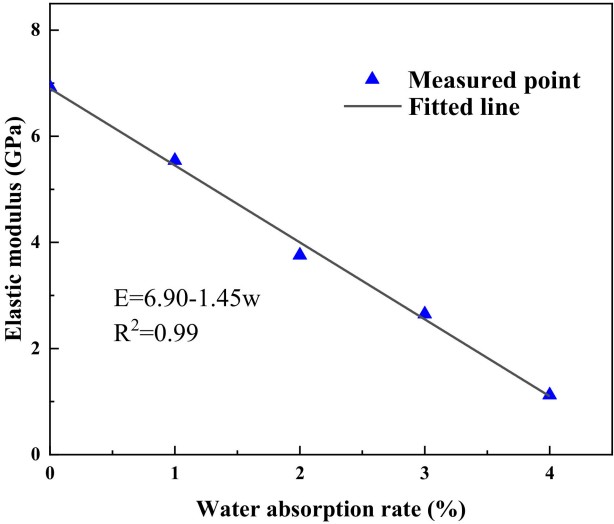

**Figure 9.** Relationship between the elastic modulus and moisture absorption rate.

In conclusion, the UCS and elastic modulus of specimens were significantly influenced by the moisture absorption rate during gaseous water sorption. Furthermore, both declined linearly with the rise of the moisture absorption rate. The calculation formulas could provide a reference for engineering design and numerical simulation studies related to red-bed mudstone in Central Sichuan, China under a high humidity environment.

(4)    Timing characteristics of UCS and elastic modulus deterioration

The time effect is a factor that cannot be ignored in the stability and safety assessment of rock masses [41]. Since the tests in Section 4.1 indicated that gaseous water sorption caused the long-term swelling of the specimen, it can be believed that the internal structure of the specimen also changed long-term during this process. In addition, the time required to prepare a specimen with a 4% moisture absorption rate was very long, taking more than two months. Therefore, to clarify the degradation characteristics of the uniaxial compression performance of mudstone during gaseous water sorption, it was not enough to obtain only the relationship between the UCS and elastic modulus and the moisture absorption rate. The time effect should also be considered.

Figure 10 demonstrated the variation of the UCS of the specimens with the moisture absorption time. After fitting the data, a good negative exponential relationship was found, and the relationship equation was shown in Equation (4) with $R^2 = 0.99$. Before 200 h, the UCS dropped quickly. After 200 h, the speed rate of its change began to descend. In particular, it had almost no significant change in time after 1000 h. This illustrated that the time-dependent behaviors of the decrease in UCS in the process of gaseous water sorption were obvious.

$$\sigma = 22.71e^{(-t/163.41)} + 6.44 \qquad (4)$$

where $\sigma$ was UCS (MPa), and t was moisture absorption time (h).

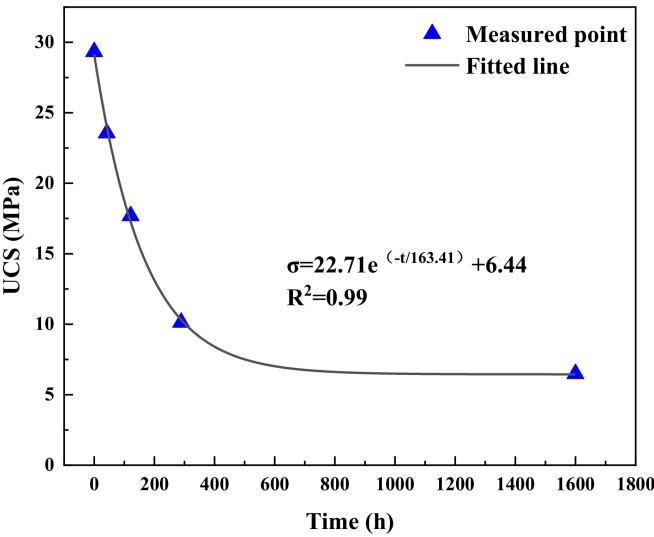

**Figure 10.** Fitting curve of the relationship between UCS and time.

Figure 11 suggested the variation of the elastic modulus of the mudstone with the moisture absorption time. After fitting the data, a good negative exponential relationship was found, and the relationship equation was displayed in Equation (5) with $R^2 = 0.97$. Moreover, similar to UCS, the deterioration of the elastic modulus also had obvious time-dependent characteristics during gaseous water sorption. The speed rate of deterioration was fast in the early stage and slow in the late.

$$E = 5.55e^{(-t/174.94)} + 1.25 \qquad (5)$$

where E was elastic modulus (GPa), and t was moisture absorption time (h).

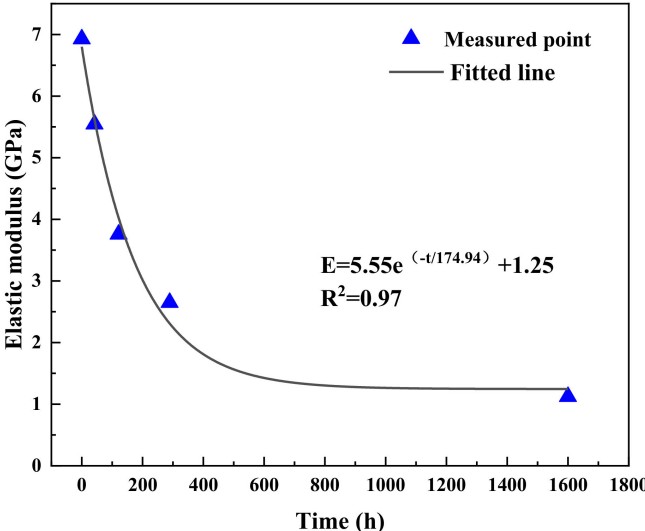

**Figure 11.** Fitting curve of relationship between elastic modulus and time.

## 5. Scanning Electron Microscopy Tests

The swelling of the mudstone in the process of gaseous water sorption indicated that gaseous water sorption must lead to a change in the microstructure of the specimens. There have been many findings on the impact of gaseous water on the macro mechanical index of rocks. However, its influence on the microstructural of mudstone was little known. In addition, in existing studies, research on microstructural played a crucial role in revealing the mechanism of liquid water sorption and swelling [14,35] and the cause of the mechanical deterioration [3] of rocks. Therefore, it was meaningful to carry out an investigation of the microstructure of mudstone. On the one hand, this could reveal the characteristics of the changes in the microstructure of mudstone during gaseous water sorption and enrich the study on the effect of gaseous water on rocks. On the other hand, it helped to explain the microscopic mechanism of the mechanical property deterioration of mudstone.

In this section, SEM tests were carried out. The microstructural changes of samples during the gaseous water sorption were studied from two perspectives, such as the morphology and structure type of particle and pore characteristics.

### 5.1. The Morphology and Structure Type of Particles

Before SEM tests, the natural mudstone block was first broken using a hammer and flake specimens with about 10 mm in length and width were selected from the small broken blocks. Then, the specimens were dried in a temperature-controlled oven. The temperature in the drying oven was set to 60 °C and the drying time was 36 h. Finally, they were placed in a closed container for gaseous water sorption. Besides, their mass changes were measured by a high precision balance for calculating the moisture absorption rate. When their moisture absorption rate reached 0%, 2%, and 4%, respectively, SEM tests were performed.

The microstructure of the specimens before and after gaseous water sorption was scanned using a Quanta 250 SEM (magnification 6~1,000,000×). Electron micrographs with a magnification of 5000× (see Figure 12) were used as examples for analysis. Figure 12 demonstrated that the aggregation form between the clay particles of the specimen before and after moisture absorption was mostly in the form of face-to-face overlap and edge-to-face overlap. Moreover, the morphology of the particles was predominantly curly and curved flakes with a few single grains. The sample before moisture absorption was denser in structure and the crack development was not obvious. In contrast, the samples after gaseous water sorption displayed many vertical cracks (see red dashed-line border) and structural deterioration.

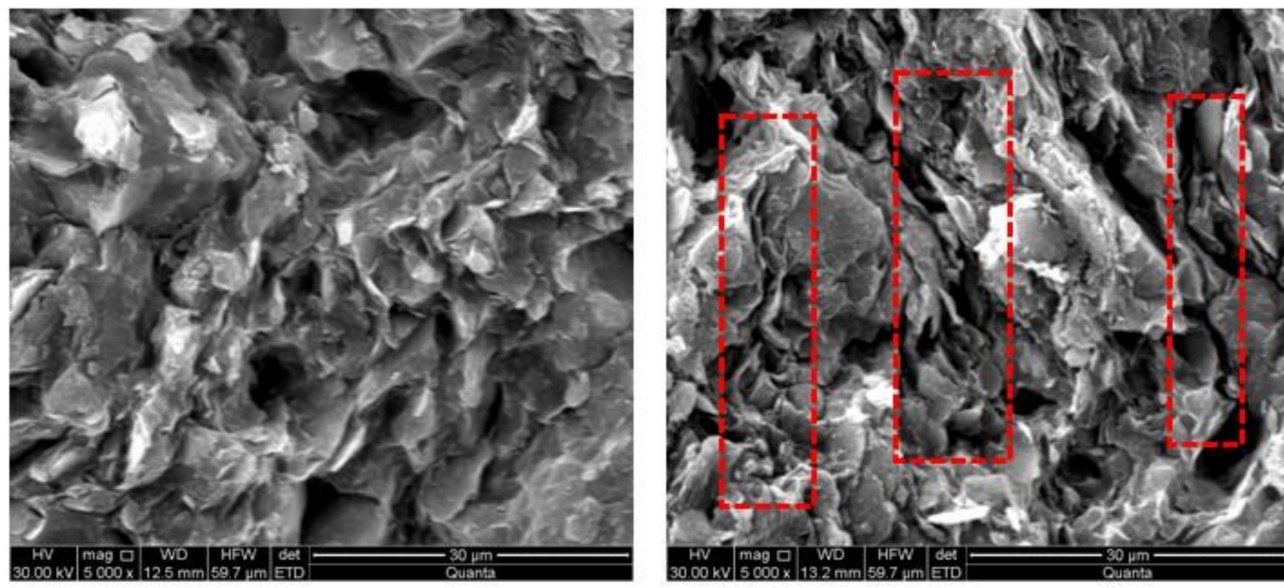

(a) Before moisture absorption    (b) After moisture absorption

**Figure 12.** Results of SEM before and after gaseous water sorption.

*5.2. Quantitative Analysis of Pore Characteristics*

Porosity and fractal dimension can describe the pore characteristics of the specimen well [42], because the former can indicate the area share of the overall pores, and the latter can quantify the complexity of the pore structure. Figure 13 illustrated the process of pore segmentation by Avizo software using the interactive thresholding method (all threshold ranges were set to 0–65. The reason was that, after many attempts, it could be found that the pores were segmented best when threshold ranges were set in this range). It also exhibited that the pores became denser and denser during moisture absorption. Moreover, they were connected with each other, forming many long and large pores.

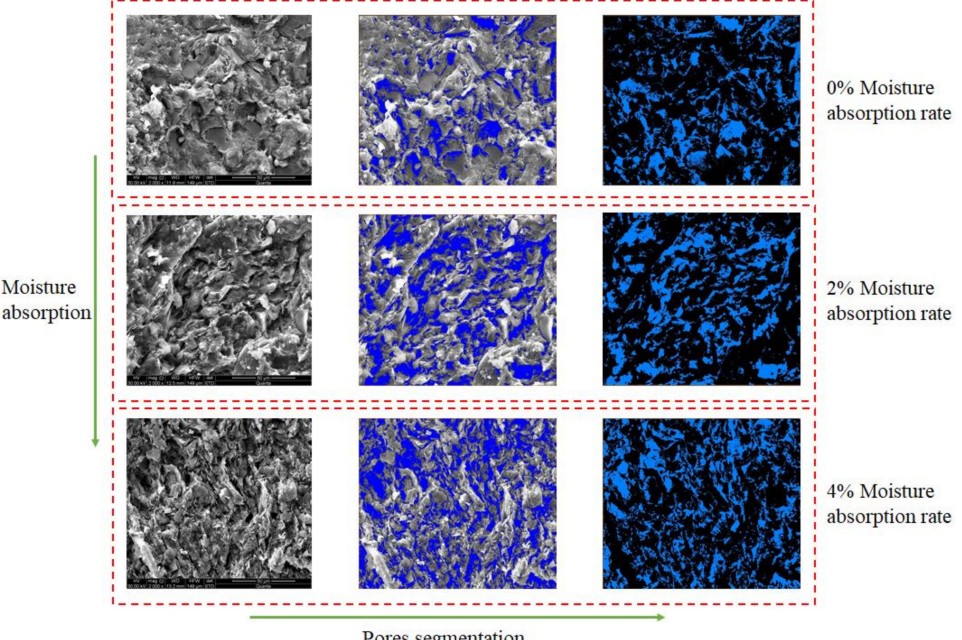

**Figure 13.** The process of pores segmentation.

"Volume Fraction" and "Fractal Dimension" commands were used to calculate the porosity and fractal dimension of the specimens, respectively. The results were shown in Table 5 below. It indicated that the porosity and fractal dimension rose with the increase in the moisture absorption rate in the process of gaseous water sorption. This was consistent with the experimental phenomenon of increasing strain with the increasing moisture absorption rate in the crack closure stage in Section 4.2. Both pointed out that from the general trend, the change of the pore structure of the mudstone was dominated by extension during gaseous water sorption. Besides, this was the microscopic reason for the decline in the UCS and elastic modulus of specimens in this process, because the expansion of the pores could destroy the structural integrity of the mudstone and cause the structure to become loose.

**Table 5.** Porosity and fractal dimension of the samples during gaseous water sorption.

| Image Magnification | Sample's State | The Number of Image | Average Porosity (%) | Average Fractal Dimension |
|---|---|---|---|---|
| 2000 | 0% moisture absorption rate | 5 | 23.27 | 1.60 |
| | 2% moisture absorption rate | 5 | 26.33 | 1.61 |
| | 4% moisture absorption rate | 5 | 29.66 | 1.67 |

To further investigate the characteristics of pores in the samples during the gaseous water sorption, the "Label Analysis" command was used to obtain the area of all pores. On this basis, as Figure 14 shows, the pore distribution was obtained with the boundaries of the pore area 0.1 $\mu m^2$, 0.5 $\mu m^2$, 1 $\mu m^2$, 5 $\mu m^2$, 10 $\mu m^2$, 50 $\mu m^2$, and 100 $\mu m^2$. The vertical coordinate of Figure 14a indicated the percentage of the number of pores (the sum of the number of pores in the corresponding area range divided by the total number of pores). It could be seen that during the gaseous water sorption, the pore composition of the samples was always dominated by tiny pores (area less than 0.1 $\mu m^2$) and the percentage of large pores (area greater than 100 $\mu m^2$) was the least. Moreover, within each range of pore area larger than 0.1 $\mu m^2$, the percentage of the number of pores of the moisture-absorbed samples was higher than that of the non-wetted samples. However, in the range of less than 0.1 $\mu m^2$, the former was only 78.7% of the latter. In brief, the percentage of tiny pores (area less than 0.1 $\mu m^2$) decreased, and the percentage of larger pores rose during the gaseous water sorption. This indicated that gaseous water sorption caused the expansion of small pores and led to their interpenetration with the surrounding neighboring pores into one larger pore.

The meaning of the vertical coordinate of Figure 14b was the percentage of the pore area (the sum of the pore area in the corresponding area range divided by the total pore area). Figure 14b displayed that regardless of before and after gaseous water sorption, large pores (area greater than 100 $\mu m^2$) contributed the most to the total pore area and tiny pores (area less than 0.1 $\mu m^2$) contributed the least. In addition, the percentage of the pore area of the moisture-adsorbed samples was higher than that of the non-absorbed samples only in the range of pore area larger than 10 $\mu m^2$ and smaller than 100 $\mu m^2$. This might be caused by small pores interpenetrating into large pores. It was noteworthy that for large pores with a pore area larger than 100 $\mu m^2$, Figure 14a indicated that the percentage of the number of pores after moisture absorption was slightly higher than that without moisture absorption. However, Figure 14b demonstrated that the percentage of the large pore area of the sample after moisture absorption was lower than that before moisture absorption. In brief, the average area of large pores (pore area larger than 100 $\mu m^2$), before moisture absorption, was larger than that after moisture absorption. This showed that gaseous water sorption led to the closure of existing large pores.

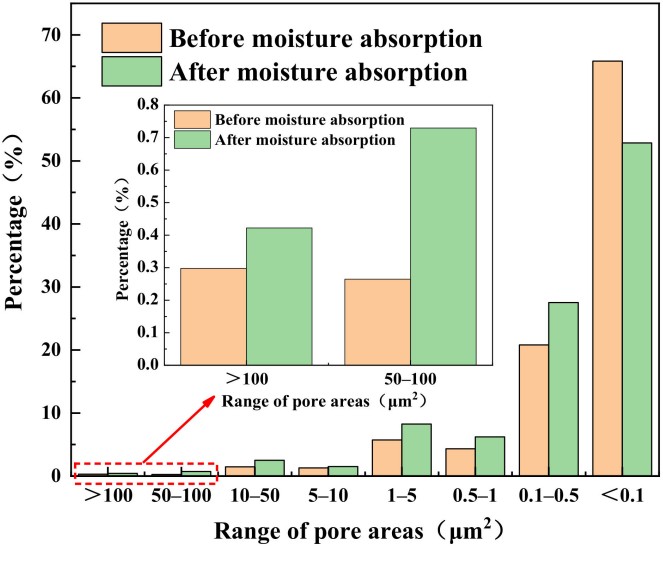

(**a**)

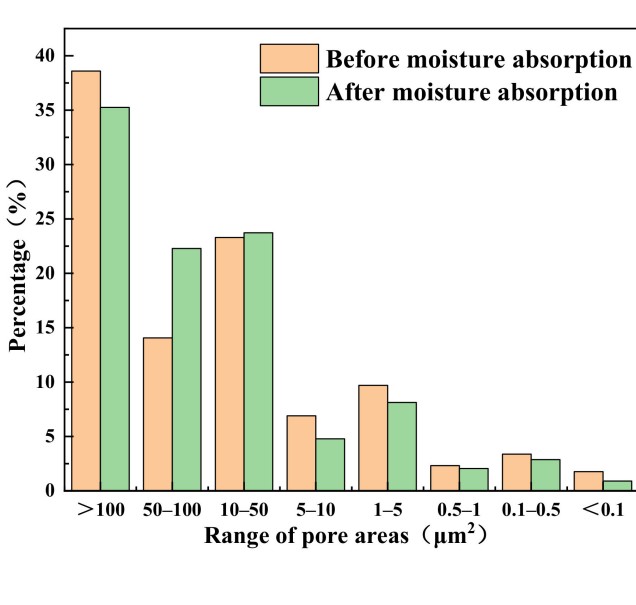

(**b**)

**Figure 14.** The distribution of pores according to the size of the area. (**a**) The vertical coordinate was the percentage of number of pores (the sum of the number of pores in the corresponding area range divided by total number of pores); (**b**) The vertical coordinate was the percentage of pore area (sum of pore areas in the corresponding area range divided by total pore area).

Consequently, although the morphology and structural types of microscopic particles did not change much in the process of gaseous water sorption, the pore structure of the specimens changed obviously. It mainly exhibited that the small pores expanded and interconnected with the surrounding pores to form larger pores. Meanwhile, the closure of the existing large pores occurred. Moreover, the porosity and fractal dimension of the mudstone increased. These were the microscopic mechanisms of the continuous softening of the specimens and the decreasing UCS and elastic modulus during gaseous water sorption.

## 6. Conclusions

Based on the above results and discussion, the following conclusions could be obtained.

(1) Gaseous water sorption could change the properties of the mudstone. Although the swelling rate of the specimen was small during gaseous water sorption, gaseous water could seriously affect its uniaxial compression performance. In this process, its softening characteristics were obvious. With the moisture absorption rate increased, the UCS and elastic modulus of mudstone decreased rapidly and linearly.

(2) Gaseous water sorption could have long-term effects on the uniaxial compression performance of mudstone. In addition, the deterioration of its UCS and elastic modulus during gaseous water sorption had obvious time-dependent characteristics. There were good negative exponential relationships between them and the moisture absorption time.

(3) Although the pores of the mudstone expanded and closed in this process, the expansion of the pores was in the dominant position and the complexity of the pore structure rose. Due to gaseous water sorption, the porosity of the specimens rose, and their structural integrity deteriorated. This was the main reason for the decrease in the uniaxial compressive properties of mudstone.

(4) It is hoped that the findings in this study can provide a meaningful reference for the design, construction, and maintenance of mudstone-related projects in high-humidity environments in central Sichuan, China.

Only a uniaxial compression test was carried out, and the influence of gaseous moisture absorption on the mechanical properties of mudstone was preliminarily discussed. In the future, triaxial compression and Brazilian disk tests on mudstone with different moisture absorption rates will be planned to reveal the effect of gaseous water sorption on the mechanical properties of mudstone more comprehensively. Moreover, when combining the research results of this paper and the existing study on the swelling of red-bed mudstone when it is completely immersed in water, it is obvious that there are large differences between the swelling laws of liquid water sorption and gaseous water sorption. Thus, this could also be a research direction.

**Author Contributions:** Conceptualization, H.Z. and Z.F.; Methodology, H.Z. and F.Y.; Software, Z.F.; Validation, Z.F. and S.L.; Formal Analysis, H.Z.; Investigation, S.L.; Resources, H.Z. and F.Y.; Data Curation, H.Z. and Z.F.; Writing—Original Draft Preparation, Z.F.; Writing—Review and Editing, H.Z. and F.Y.; Visualization, Z.F.; Supervision, H.Z.; Project Administration, H.Z.; Funding Acquisition, H.Z. All authors have read and agreed to the published version of the manuscript.

**Funding:** This study was funded by the National Natural Science Foundation of China (grant number 52178182).

**Institutional Review Board Statement:** Not applicable.

**Informed Consent Statement:** Not applicable.

**Data Availability Statement:** The data presented in this study are available on request from the corresponding author.

**Conflicts of Interest:** The authors declare no conflict of interest.

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
