# Peer review of "Study on Deterioration Characteristics of Uniaxial Compression Performance and Microstructure Changes of Red-Bed Mudstone during Gaseous Water Sorption"

_buildings, doi:10.3390/buildings12091399_

Round 1
Reviewer 2 Report
Red-bed mudstone is prone to collapse under the influence of water, and has geological characteristics such as serious strength attenuation, strong hydrophilicity and weak water permeability, which have caused many engineering problems worldwide. This makes the study of the influence of water on mudstone become a hot topic in engineering construction. Aiming at the deterioration of uniaxial compression performance in the process of gas water absorption, the authors carried out expansion test, uniaxial compression test and scanning electron microscope (SEM) to clarify the variation characteristics of mudstone when it is exposed to water. The thesis has a good topic selection, and the research conclusion has certain theoretical value and engineering practicability. The writing logic of the paper is good. In order to further improve the research results, the following problems still need to be further improved. Specific questions are as follows:
(1) The introduction part of this paper only lists the research status, but does not summarize the research status, and does not reflect the innovation of this paper.
(2) Some pictures in the paper do not match the description of the text, for example: "The description of the 5000x scanning electron microscope picture is labeled as Figure 10", "the description of the pore segmentation picture is labeled as Figure 11".
(3) During the preparation of rock samples, did the author take into account the homogeneity of water absorption of rock samples, whether there was any difference between the water absorption rate on the surface of rock samples and the water absorption rate inside of rock samples, and what methods were used to solve this difference?
(4) The residual strength of rock after uniaxial compression peak is an important physical and mechanical property to describe the damage and deterioration of rock. Have the authors considered the relationship between the characteristics of red-bed mudstone after failure and the water absorption of rock?
(5) In this paper, the SEM images were processed by pore segmentation, and the reason for selecting the threshold range (0-65) was not explained, so there was no basis for selecting the threshold.
Reviewer 3 Report
The manuscript "Study on Deterioration Characteristics of Uniaxial Compression Performance and Microstructure Changes of Red-Bed Mudstone during Gaseous Water Sorption" by Hongbing Zhu, Zhenghao Fu, Fei Yu3, Sai Li was submitted for review.
I read the submitted manuscript with great interest. The author turned to a very urgent problem: study of moisture on changes in the properties of mudstone. The authors have done a very good study, but the manuscript has several significant flaws that need to be corrected. Correction of the shortcomings listed below must be done to improve the quality of the manuscript, enhance the ease of perception of the presented material and increase the interest of a readers.
1.) From my point of view, there are very few keywords. Keywords enable the reader to quickly search for the necessary material and enable the author to popularise their research and increase interest and citations. But if this number of keywords satisfies the requirement of the journal, this comment is advisory.
2.) The abstract is not quite formed correctly. It is very blurry and framed incorrectly. The abstract should clearly indicate the purpose of the study, its importance for society (i.e. to characterize the problem), identify the methods and materials of the study, and the conclusions should be clearly and briefly formulated. It seems that the authors have taken certain phrases from the text and thus formed the abstract. There is no "starting point" in the abstract, that is, information about previous studies (one sentence is enough). From my point of view, in the abstract, such information begins with the statement: "Previously conducted studies have established that ...".
2.1) It is desirable to avoid narrative text in the abstract.
2.2) Try to use words and phrases: an analysis has been carried out; studied; developed; proposed; established and so on. It is advisable to start sentences in the abstract with these words and phrases.
2.3) At the end of the abstract, it is necessary to indicate the final result obtained by the authors, for example: A model has been developed that allows ...; A dependence has been established which is...; A pattern has been revealed...; An efficient system (technology) has been proposed, and so on.
2.4.) The abstract should briefly display the methods.
The abstract should be revised.
3.) The manuscript has a very meager list of references in terms of geography. Very weak geography of citations. The list of references is intended to demonstrate the depth of the author's study of the material, the relevance and interest of their research.
3.1.) The depth of study is demonstrated with the number of references (34 references) - on the verge of sufficiency, preferably more.
3.2.) Relevance – with the availability of research in recent years – there is about 33% of papers in the last five years - on the verge of sufficiency.
3.3.) Interest – with the availability of research by scientists from different countries - is not enough. There are not references to international studies. Since you are publishing your manuscript in an international publication, it is necessary to demonstrate the international relevance and interest of this issue. This can be done by analyzing the studies of scientists from different countries. It is imperative to supplement the list of references with studies of scientists from different countries to show interest and relevance.
The List of References needs to be completed.
4.) From my point of view, in the introduction, when analyzing previous studies, the authors greatly abuse the mention of surnames. The authors try to accompany each statement with proof (reference). It is very good. But in the list of references, there are already the names of the scientists who performed the study. Mentioning names in the text is redundant. It is important for the reader to know the essence (main idea) of the research you are referring to, not the name of the researcher.
5.) At the end of the introduction, there is no conclusion on the analysis carried out. This conclusion allows to characterize the actual question posed, the purpose of the study and the tasks to be solved to achieve this goal. For example: Analyzing the above, it can be noted that ... is a very topical issue. Therefore, the purpose of this study is ... and to achieve this, it is necessary to solve the following tasks: 1); 2); ... Such a conclusion at the end of the introduction allows the researchers to properly formulate the conclusions of the study and the reader to understand the vector of the study.
6.) It is not clear to me from the introduction, and I think readers will also find it difficult to guess the ultimate purpose of the study. The authors in their manuscript investigate the effects of moisture on changes in mudstone properties. Such a study is done for something. Simply investigating the properties of mudstone is a dead-end branch of research. The properties of mudstone are needed for something. In my view, this needs to be noted in the introduction.
7.) Regarding remark (6), I would like to point out that the authors have very poorly disclosed the main subject of the study. The authors did not specify why mudstone was chosen for the study exactly.
In the introduction the authors did not describe the problems arising from mudstone moistening and why they are dangerous.
In recent years a lot of work has been carried out on the study of soils in the area of construction work and the rock mass in the area of mining. For example:
7.1) Dobrzycki, P.; Kongar-Syuryun, C.; Khairutdinov, А. Vibration reduction techniques for Rapid Impulse Compaction (RIC). J. Phys.: Conf. Ser. 2020, 1425(1), 012202. https://doi.org/10.1088/1742-6596/1425/1/012202. The authors investigated soils to establish the radius of propagation of vibrations, the possibility of reducing them depending on the density of the soil.
7.2) Golik, V.I.; Kongar-Syuryun, Ch.B.; Michałek, A.; Pires, P.; Rybak, A. Ground transmitted vibrations in course of innovative vinyl sheet piles driving. J. Phys.: Conf. Ser. 2021, 1921(1), 012083. https://doi.org/10.1088/1742-6596/1921/1/012083. The subject of the study is also various soils, which have been studied to establish the relationship between the depth of the innovative vinyl sheet piles driving and the vibrations transmitted by the soil depending on its density.
7.3) Rybak, J.; Khayrutdinov, M.M.; Kuziev, D.A.; Kongar-Syuryun, Ch.B.; Babyr, N.V. Prediction of the geomechanical state of the rock mass when mining salt deposits with stowing. Journal of Mining Institute. 2022, 253, 61-70. https://doi.org/10.31897/PMI.2022.2. In the study, the authors predict the rock mass behavior during salt mining depending on the strength of the pillars and the strength characteristics of the stowing.
As follows from the presented works (7.1) - (7.3) the authors of the manuscript submitted for review missed a large layer of research related to soils and rock mass studies. It is also demonstrated by this survey that any study of the soil or rock mass has a definite end goal. If the authors become familiar with the works presented in (7.1), (7.2), (7.3) they will be able to properly form the introduction, enrich their manuscript with international research by scientists from Poland, Slovenia, Slovakia, Serbia and Russia, and demonstrate the depth of their material, as well as eliminate the remark (3.3).
8.) The authors describe the testing procedure in section 3. However, not all necessary methods are described in detail, which does not allow readers to understand the experiment and, if necessary, to repeat it.
8.1) Lines 158-161. The authors refer to mudstone drying. However, the drying procedure is not fully described. What equipment was used for drying, how long did it take to dry, how or on what equipment did the authors establish that there was no moisture in the material under study.
8.2) Line 162. By which method is it established that no obvious cracks appeared on the surface of the samples?
8.3) Lines 228-229. The authors state: "Therefore, when the specimen had less original pores, less moisture absorption and less swelling might occur". However, as far as I understand, the authors did not examine the samples for the presence of pore spaces in them and did not conduct studies on the absorption of moisture by the samples, depending on the difference in voids. Hence, what was the basis for that conclusion.
8.4) It is necessary to indicate the equipment on which the samples were tested for uniaxial compression.
8.5) It is necessary to indicate on what (on which element of the sample) the microstructural analysis was carried out.
9.) From my point of view, the conclusions are completely formed incorrectly. There is a lot of redundant information in the conclusions related to other sections. For example: lines 471-474 refer more to "methods". It is redundant for conclusions. Conclusions should briefly characterize the result of the study, for example:
As a result of the study
(1) the dependence of … was obtained.
(2) it was found that ...
(3) and so on.
The conclusions presented by the authors are overloaded with information that they have previously presented in methods and results. Such a presentation reduces the ease of perception by the reader of the information presented. The conclusions need to be redone.
Summary: The manuscript is not a finished research work. Corrections are needed. Despite the rather impressive list of comments, the authors carried out a good research work. The scientific component of the manuscript is really of interest to the readers. From my point of view, the authors failed to present their research correctly and clearly, which reduced its value and worsened the ease of perception of the material presented.
From my point of view, the manuscript is worthy of publication in the open press with corrections in accordance with my suggestions.
Round 2
Reviewer 3 Report
The manuscript «Study on Deterioration Characteristics of Uniaxial Compression Performance and Microstructure Changes of Red-Bed Mudstone during Gaseous Water Sorption» by Hongbing Zhu, Zhenghao Fu, Fei Yu, Sai Li was submitted for second review.
As can be seen from the submitted manuscript and the explanatory note to the review, the authors did a lot of work to make changes in accordance with the comments.
The revised manuscript is a completed scientific study on a highly relevant topic: study of moisture on changes in the properties of mudstone. The revised version of the manuscript, in my opinion, fully satisfies the requirements of a scientific article and can be published in the open press.
